materials science

rubberwood, small-diameter log, spindleless lathe technology, veneer thickness, pressing temperature, gluebond shear strength

**Authors for correspondence:**
K. L. Chin
e-mail: kitling.chin419@gmail.com
P. S. H'ng
e-mail: ngpaiksan@gmail.com

# Physical properties and bonding quality of laminated veneer lumber produced with veneers peeled from small-diameter rubberwood logs

P. S. Khoo[1], K. L. Chin[1], P. S. H'ng[1,2], E. S. Bakar[2], C. L. Lee[1], W. Z. Go[2] and R. Dahali[2]

[1]Institute of Tropical Forestry and Forest Products, and [2]Faculty of Forestry, Universiti Putra Malaysia, 43400 UPM Serdang, Selangor, Malaysia

PSK, 0000-0002-8584-9198; PSH, 0000-0002-3641-7698; CLL, 0000-0002-2293-4088

The peeling of small-diameter rubberwood logs from the current short-rotation practices undoubtedly will produce lower grade veneers compared to the veneers from conventional planting rotation. Hence, this raises the question of the properties of the produced laminated veneer lumber (LVL) from veneers peeled from small-diameter rubberwood logs using the spindleless lathe technology. Different thicknesses of rubberwood veneers was peeled from rubberwood logs with diameter less than 20 cm using a spindleless lathe. Three-layer LVLs were prepared using phenol formaldehyde (PF) adhesive and hot pressed at different temperatures. During the peeling of veneer, lathe checks as deep as 30–60% of the veneer thickness are formed. Owing to deeper lathe check on 3 mm rubberwood veneer, higher pressing temperature significantly increased the gluebond shear strength of the PF-bonded LVL. In addition, lathe check frequency was also shown to influence the bond strength. The presence of higher lathe check frequency on 2 mm veneer increased the wettability, thus facilitating optimum penetration of adhesive for stronger bonding. These findings stress the importance of measuring and considering the lathe check depth and frequency during the lamination process to get a better understanding of bonding quality in veneer-based products.

# 1. Introduction

Rubberwood is a well-known species and highly requested by the wood industry particularly for biocomposite production [1]. However, there has been a short supply of rubberwood logs for veneer production owing to the inadequate diameter of the logs for veneer peeling. Instead of the conventional planting rotation of 25–30 years, nowadays, rubber plantations in Malaysia are managed under a short planting cycle and the trees will be probably felled when they are around 15 years owing to the high demand for latex and rubberwood [2], which resulted in harvesting small-diameter logs. The invention of a spindleless lathe in recent years had encouraged the use of fast-growing plantation species with small-diameter logs in the production of structural composite lumber such as laminated veneer lumber (LVL) [3].

The mechanical properties of LVL generally depend on a variety of factors particularly veneer quality (moisture content, density, lathe checks, surface roughness and thickness variation), adhesive quality (type of adhesive, mixture of resin and viscosity) and bonding quality (wettability, glue spread rate, pressure, time and temperature) [4,5]. Numerous factors contribute to the veneer quality. The most common factors are owing to the log's characteristics (species, log diameter, log quality, specific gravity, wood pores, grain structure, juvenile and mature wood) [4].

Owing to younger age and shorter planting cycle, the juvenile wood is proportionally higher than the mature wood [6]. This higher proportion of juvenile wood can have a significant effect on the log quality and the veneer quality [6,7]. The presence of huge proportion of juvenile wood in a log will cause excessive shrinking and swelling, problems of warping, fuzzy grain and general instability in the manufacture and the use of the wood. These problems may occur in wood after sawing, veneering, drying and machining [8]. Darmawan *et al.* [4] revealed that veneer with severe lathe checks will be produced when fast-growing species with a higher proportion of juvenile wood are peeled. Peeling veneers from small-diameter logs produced deeper lathe checks, longer lathe checks and larger interval between lathe checks [2,9,10]. Veneer produced from small-diameter rubberwood logs using spindleless lathes undoubtedly have very different properties compared with veneers produced from large-diameter logs using conventional spindled lathes.

To date, however, to our knowledge there has been no published report on the production of LVL with veneers peeled from small-diameter rubberwood logs using a spindleless lathe. Owing to the unique properties of veneer peeled from small-diameter rubberwood logs, the optimum parameters required for the production of LVL have to be determined. Therefore, the purpose of this research was to evaluate the physical properties and gluebond shear strength of rubberwood LVL manufactured from different veneer thicknesses at different pressing temperatures. The LVL properties are important in determining the potential of LVL produced with veneers peeled from small-diameter rubberwood logs using spindleless lathe technology.

# 2. Material and methods

Small-diameter rubberwood logs (between 15 and 18 cm) were peeled according to the method demonstrated by Khoo *et al.* [2] to produce veneer thicknesses of 1, 2 and 3 mm, and the veneer properties were listed in table 1.

Three-layer LVL specimens were prepared from 1, 2 and 3 mm thickness rotary peeled rubberwood veneers with the moisture content of $8 \pm 2\%$ [11]. Phenol formaldehyde (PF) adhesive with 45% solid content was obtained from Aica Chemicals (M) Sdn. Bhd. Properties of the PF adhesive were as follows: specific gravity of 1.232 at 30°C, pH of 12.90 at 30°C, viscosity of 69 Cps at 30°C and gel time of 21 min at 105°C. Commercial filler was used with the PF adhesive. Double glueline spread rate of $200 \, \mathrm{g \, m^{-2}}$ was applied on the veneer surface; the veneer sheets were pressed together parallel to each other and the loose side of the veneer was placed towards the centre of the boards. The three plies of LVLs were hot pressed at 120, 140 and 160°C for 5 min with $7 \, \mathrm{kgf \, cm^{-2}}$ specific pressing pressure. Each treatment was carried out in three replicates. After hot pressing, the LVLs were conditioned until they reached the equilibrium moisture content.

## 2.1. Evaluation

### 2.1.1. Moisture content and specific gravity

The moisture content of the produced LVL was determined using the conventional drying method according to the ASTM D 4442-03. The specimens were oven dried at $103 \pm 2°C$ until the constant

**Table 1.** Properties of rotary peeled rubberwood veneers obtained by using a spindleless lathe [2].

| veneer thickness (mm) | lathe check properties | | contact angle (°C) after 10 s |
| --- | --- | --- | --- |
| | depth (%) | frequency per 5 cm | |
| 1 | 30 ± 10 | 35 ± 10 | 15 |
| 2 | 50 ± 15 | 30 ± 10 | 8 |
| 3 | 60 ± 10 | 25 ± 5 | 34 |

weight is obtained to determine the oven-dry weight. The moisture content of the specimens was calculated as follows:

$$\text{moisture content}\,(\%) = \frac{\text{initial weight (g)} - \text{oven} - \text{dry weight (g)}}{\text{oven} - \text{dry weight (g)}} \times 100\%.$$

The specific gravity of test samples with dimensions of 50 mm by 50 mm was determined according to the ASTM D 2395-02. Oven-dry density and specific gravity of the specimens were calculated using the following equations:

$$\text{oven} - \text{dry density}\,\left(\frac{\text{g}}{\text{cm}^3}\right) = \frac{\text{oven} - \text{dry mass}}{\text{oven} - \text{dry volume}}$$

and

$$\text{specific gravity} = \frac{\text{oven} - \text{dry density}}{\text{density of water}}.$$

### 2.1.2. Water absorption and volumetric swelling

Test specimens with dimensions of 50 mm by 50 mm were weighed, and the radial (thickness), tangential and longitudinal directions were measured before being submerged in 25 mm of distilled water maintained at a temperature of 20 ± 1°C. After a 2 h of submersion, the water was removed and the specimens were drained for 10 ± 2 min to remove excess surface water. The specimens were weighed, and the radial, tangential and longitudinal of the specimens was measured immediately. After that, the specimens were submerged for an additional 22 h and followed by the weighing and measuring procedures mentioned earlier. After submersion, the specimens were put in oven at 103 ± 2°C to calculate the moisture content based on oven-dry weight. Based on the ASTM D 1037-12, the percentages of water absorption, radial, tangential, longitudinal and volumetric swelling were determined using the following equations:

$$\text{water absorption}\,(\%) = \frac{\text{final weight (g)} - \text{initial weight (g)}}{\text{initial weight (g)}} \times 100\%,$$

$$\text{swelling}\,(\%) = \frac{\text{final length (mm)} - \text{initial length (mm)}}{\text{initial length (mm)}} \times 100\%$$

and

$$\text{volumetric swelling}\,(\%) = \frac{\text{final volume (cm}^3\text{)} - \text{initial volume (cm}^3\text{)}}{\text{initial volume (cm}^3\text{)}}.$$

### 2.1.3. Gluebond shear strength

For gluebond shear strength, the cutting pattern for testing specimens is shown in figure 1.

Testing specimens were tested according to the ASTM D 906 using an INSTRON Universal Testing Machine. The load was applied continuously throughout the test at a uniform rate of motion of the movable crosshead of the testing machine of 4 mm min$^{-1}$. The shear strength of each specimen was calculated from the following equation:

$$\text{shear strength} = \frac{F}{l \times b},$$

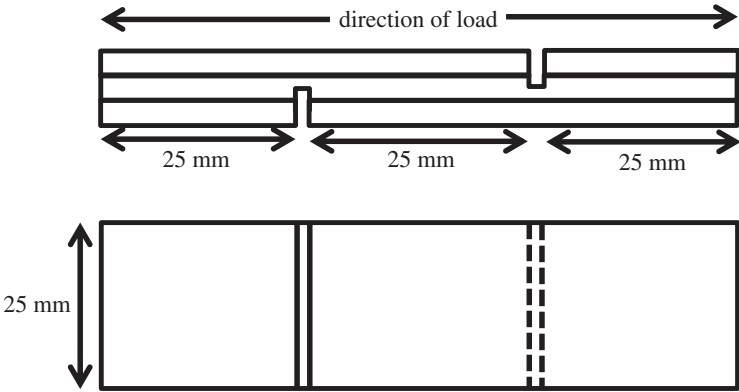

**Figure 1.** Cutting pattern of gluebond shear test specimen.

where $F$ is the failing force of the specimen, in Newtons; $l$ is the length of the shear area, in mm; and $b$ is the width of the shear area, in mm.

The glue penetration of the rubberwood veneer was examined using an Olympus SZX12 stereo microscope to evaluate the interaction between the adhesive and rubberwood veneers. Specimens were taken at the cross section of each LVL panel. The LVL specimens of size 10 mm wide × 10 mm long were cut for assessment. The specimen was examined under microscope at 75× magnification.

# 3. Results and discussion

As shown in table 3, the interaction between veneer thickness and pressing temperature was analysed using a one-way analysis of variance (ANOVA). The interaction of veneer thickness and pressing temperature has a highly significant effect on the physical properties and gluebond shear strength of three plies of rubberwood LVLs. The significant mean values were compared using Tukey's test, and the results are tabulated in table 3.

## 3.1. Moisture content and specific gravity

After conditioning for two weeks, the equilibrium moisture content of LVLs produced from different thickness of rubberwood veneers were $13 \pm 2\%$. Based on the verdicts, there was an increase in the density of rubberwood LVLs compared to the solid rubberwood. Air dry density of solid rubberwood generally ranges from 650 to 706 kg m$^{-3}$ according to the study by Khoo *et al.* [2]. In this study, the air dry density of rubberwood LVLs ranged from 736 to 857 kg m$^{-3}$, which was higher than the density of solid rubberwood. The increase in the density of 3-ply rubberwood LVL is probably owing to addition of adhesive. The density of PF resin used in this study is 1.2 g cm$^{-3}$, which is much higher than that of the substrate. Hence, the total mass of panels was greatly affected by the density of the adhesive [12].

One-way ANOVA (table 3) clearly shows that the specific gravity of rubberwood LVLs is highly significantly influenced by the interaction between veneer thickness and pressing temperature. Highest specific gravity can be found in LVL produced with 1 mm veneer thickness at 120°C, whereas the lowest specific gravity rubberwood LVLs were produced using 3 mm veneer thickness at 140°C. Based on table 2, the effect of pressing temperature was highly significant on the specific gravity of LVL. With the increasing pressing temperature, reduction in specific gravity was more prominent in rubberwood LVL produced with 1 mm veneer thickness. Reduction in specific gravity results in the degradation and changes in both chemical and physical properties owing to the exposure of wood to the higher temperature [6]. Between 100 and 200°C, dehydration and decarboxylation processes happen and wood generates water vapour, carbon dioxide and volatile organic compounds; thus, the carbohydrate polymers in wood depolymerize and degrade slowly as the temperature increase [13].

Regardless of the pressing temperature, LVL produced with 3 mm veneer thickness has significantly lower specific gravity compared with LVL produced with 1 and 2 mm veneer thicknesses. This variation is mainly attributed to the presence of more pores and void volume in thick veneer and resulted in lower specific gravity. High specific gravity in LVL produced with thin veneer is generally owing to the presence of less pores and void volume [14].

**Table 2.** Analysis of variance of the effect of veneer thickness and pressing temperature on the physical properties and gluebond shear strength of rubberwood LVLs. (Specific gravity (SG), water absorption after 2 h immersion (WA2), water absorption after 24 h immersion (WA24), radial swelling after 2 h immersion (RS2), radial swelling after 24 h immersion (RS24), longitudinal swelling after 2 h immersion (LS2), longitudinal swelling after 24 h immersion (LS24), tangential swelling after 2 h immersion (TS2), tangential swelling after 24 h immersion (TS24), volumetric swelling after 2 h immersion (VS2), volumetric swelling after 24 h immersion (VS24) and gluebond shear strength (GBS). n.s., not significant. **Significant at $p < 0.01$.)

| properties | Pr > F | | | | | | | | | | | |
|---|---|---|---|---|---|---|---|---|---|---|---|---|
| | SG | WA2 | WA24 | RS2 | RS24 | LS2 | LS24 | TS2 | TS24 | VS2 | VS24 | GBS |
| thickness | ** | ** | ** | ** | ** | ** | ** | ** | ** | ** | ** | ** |
| temperature | ** | ** | ** | ** | n.s. | ** | ** | ** | ** | ** | n.s. | ** |
| thickness × temperature | ** | ** | ** | ** | ** | ** | ** | ** | ** | ** | ** | ** |

## 3.2. Water absorption and volumetric swelling

The interaction between veneer thickness and pressing temperature has highly significant effect on the percentage of water absorption, radial swelling, longitudinal swelling, tangential and volumetric swelling after 2 and 24 h of water immersion (table 3). Within the first 2 h immersion, rubberwood LVL pressed at higher temperature has a significantly higher percentage of radial swelling compared to others. Pressing LVL at high temperature softens the wood and leads to compression of the wood [15]. With the increasing pressing temperature, the compression of wood increases [16]. However, these compression stresses building up during hot pressing are released (springback) when panels are immersed in water [17], which resulted in higher radial swelling.

After 24 h immersion, rubberwood LVL produced from 3 mm veneer thickness at 140°C has the highest value in water absorption. This might be owing to more porosity in 3 mm veneer thickness compared with 1 and 2 mm veneer thickness. This result was supported by the specific gravity result, which mentioned that the rubberwood LVL produced from 3 mm veneer thickness at 140°C has lowest specific gravity. Because the volume of the lumina increases with a decrease in specific gravity, the void volume in cell lumina will be replaced by free water until the maximum moisture content is achieved [18]. This resulted in the highest percentage of water absorption after the sample was immersed for 24 h. On the other hand, the lowest percentage of water absorption after 24 h immersion was significantly obtained by LVL produced from 1 and 2 mm veneer thicknesses at 120°C, which are 46.46 and 45.98%, respectively. This result was in agreement with the specific gravity result, which shows that as the specific gravity increases, the volume of the lumina decreases. This decreases the maximum moisture content because less void is available for free water [18].

Based on table 3, swelling in the longitudinal direction was $0.4 \pm 0.2\%$ and $1.1 \pm 0.5\%$ after 2 and 24 h immersion, respectively. In fact, there is no longitudinal swelling in most species as stated by Uysal [19]. However, rubberwood is expected to have higher longitudinal swelling compared with other species owing to presence of tension wood [20]. Sik *et al*. [21] reported that longitudinal shrinkage in rubberwood contains tension wood up to 0.67% compared with normal shrinkage, which is about 0.1–0.3%. In this study, the longitudinal swelling of rubberwood LVL after 2 h immersion ($0.4 \pm 0.2\%$) is considerably lower compared with the longitudinal shrinkage in rubberwood containing tension wood (0.67%). Rubberwood LVL produced by thin veneer has a relatively lower longitudinal swelling after 24 h immersion compared with that produced by thicker veneer. By converting rubberwood into LVL, tension wood will disperse more uniformly in thin veneer, and it may cause a minor impact to the longitudinal swelling. With the shorter planting cycle, harvested rubberwood logs are certainly smaller in diameter and lower in wood quality owing to the presence of a higher proportion of the juvenile wood [2,6,7]. The juvenile wood has larger microfibril angle in fibres, which results in greater longitudinal shrinkage and lesser tangential shrinkage [6,7].

LVL produced by 3 mm veneer thickness was found to be the most dimensional stable panel with the least radial swelling, tangential swelling and volumetric swelling after 2 and 24 h. Nevertheless, LVL with 3 mm veneer thickness has significant increment in the percentage of water absorption from 2 h immersion to 24 h immersion. This increment owing to the LVL had reached the fibre saturation point within the first 2 h of immersion. As hydroxyl groups in cell wall polymers have been saturated with water molecules, the cell wall no longer expands to accommodate the water molecules [22]. Wood is dimensionally unstable within the first 2 h immersion owing to the moisture content below the fibre saturation point (usually 30% moisture content). Once the moisture content is beyond the fibre saturation point, the excess free water only fills lumens and makes wood heavier, but does not contribute to further expansion [19].

## 3.3. Gluebond shear strength

In terms of gluebond shear strength, the effect of veneer thickness and pressing temperature interaction was highly significant. Not only the information regarding the gluebond shear strength is crucial but also the per cent wood failure is important for the evaluation of bonding quality. The LVL failures after the gluebond shear test were observed and evaluated visually. The result is shown in figure 2.

According to table 3 and figure 2, the highest gluebond shear strength and the per cent wood failure were observed in LVL produced with 2 mm veneer thickness at 120°C, whereas the lowest gluebond shear strength and the per cent wood failure were obtained by LVL produced with 3 mm veneer thickness at 120°C. The previous study by Khoo *et al*. [2] revealed that thicker veneer usually contained deeper lathe check depth and longer lathe check length but lower lathe check frequency.

**Table 3.** Mean values of interactions between veneer thickness and pressing temperature on the physical properties and gluebond shear strength of rubberwood LVLs. (The mean value with the same letter within the same column is not significantly different $p > 0.05$. Values in parentheses indicate standard deviation.)

| thickness (mm) | temperature (°C) | properties | | | | | |
|---|---|---|---|---|---|---|---|
| | | SG | WA2 (%) | WA24 (%) | RS2 (%) | RS24 (%) | LS2 (%) |
| 1 | 120 | 0.776[e] | 29.28[ab] | 46.46[a] | 2.413[b] | 5.297[d] | 0.502[de] |
| | | (0.02) | (0.87) | (1.44) | (0.15) | (0.17) | (0.02) |
| | 140 | 0.710[cd] | 34.66[cd] | 51.14[bc] | 3.109[c] | 5.619[d] | 0.259[a] |
| | | (0.03) | (2.52) | (3.74) | (0.28) | (0.32) | (0.02) |
| | 160 | 0.674[bc] | 35.82[d] | 48.29[ab] | 2.951[c] | 5.654[d] | 0.444[c] |
| | | (0.04) | (2.90) | (3.61) | (0.08) | (0.33) | (0.03) |
| 2 | 120 | 0.737[de] | 26.11[a] | 45.98[a] | 1.594[a] | 3.648[bc] | 0.483[cde] |
| | | (0.02) | (2.18) | (2.33) | (0.09) | (0.23) | (0.04) |
| | 140 | 0.719[d] | 29.62[ab] | 47.08[ab] | 2.113[b] | 3.722[bc] | 0.367[b] |
| | | (0.02) | (2.24) | (0.72) | (0.20) | (0.26) | (0.03) |
| | 160 | 0.707[cd] | 32.24[bcd] | 47.49[ab] | 2.155[b] | 3.281[ab] | 0.525[e] |
| | | (0.03) | (2.93) | (2.15) | (0.15) | (0.20) | (0.04) |
| 3 | 120 | 0.664[ab] | 32.49[bcd] | 51.14[bc] | 1.449[a] | 3.035[a] | 0.372[b] |
| | | (0.02) | (3.19) | (3.86) | (0.14) | (0.17) | (0.03) |
| | 140 | 0.631[a] | 34.86[cd] | 54.02[c] | 1.738[a] | 3.126[a] | 0.470[cde] |
| | | (0.02) | (2.50) | (2.53) | (0.11) | (0.06) | (0.02) |
| | 160 | 0.667[ab] | 31.33[bc] | 49.58[abc] | 2.392[b] | 3.991[c] | 0.463[cd] |
| | | (0.02) | (2.92) | (1.73) | (0.11) | (0.19) | (0.03) |

| thickness (mm) | temperature (°C) | properties | | | | | |
|---|---|---|---|---|---|---|---|
| | | LS24 (%) | TS2 (%) | TS24 (%) | VS2 (%) | VS24 (%) | GBS (MPa) |
| 1 | 120 | 0.863[b] | 2.269[b] | 4.212[de] | 5.128[cd] | 10.591[de] | 3.10[ab] |
| | | (0.08) | (0.21) | (0.40) | (0.21) | (0.65) | (0.04) |
| | 140 | 1.002[bc] | 2.796[c] | 4.266[de] | 6.868[f] | 11.442[e] | 4.14[cd] |
| | | (0.09) | (0.18) | (0.15) | (0.16) | (0.32) | (0.06) |
| | 160 | 0.602[a] | 2.291[b] | 3.599[b] | 5.933[e] | 10.329[cd] | 3.56[bc] |
| | | (0.04) | (0.19) | (0.02) | (0.39) | (0.52) | (0.13) |
| 2 | 120 | 1.261[ef] | 2.200[ab] | 4.588[e] | 4.322[b] | 9.472[bc] | 5.91[f] |
| | | (0.07) | (0.14) | (0.15) | (0.28) | (0.62) | (0.35) |
| | 140 | 1.552[g] | 2.307[b] | 4.115[cd] | 5.029[cd] | 8.293[a] | 3.88[cd] |
| | | (0.14) | (0.22) | (0.21) | (0.49) | (0.47) | (0.29) |
| | 160 | 0.626[a] | 2.579[bc] | 3.951[bcd] | 5.174[d] | 8.595[ab] | 4.31[d] |
| | | (0.01) | (0.16) | (0.22) | (0.07) | (0.30) | (0.30) |
| 3 | 120 | 1.196[de] | 1.869[a] | 3.722[bc] | 3.601[a] | 8.078[a] | 2.58[a] |
| | | (0.01) | (0.13) | (0.15) | (0.27) | (0.48) | (0.18) |
| | 140 | 1.410[fg] | 2.558[bc] | 3.873[bcd] | 4.537[bc] | 8.459[a] | 3.74[cd] |
| | | (0.12) | (0.22) | (0.18) | (0.35) | (0.55) | (0.17) |
| | 160 | 1.064[cd] | 1.841[a] | 3.010[a] | 4.361[b] | 8.358[a] | 5.05[e] |
| | | (0.10) | (0.15) | (0.15) | (0.31) | (0.20) | (0.13) |

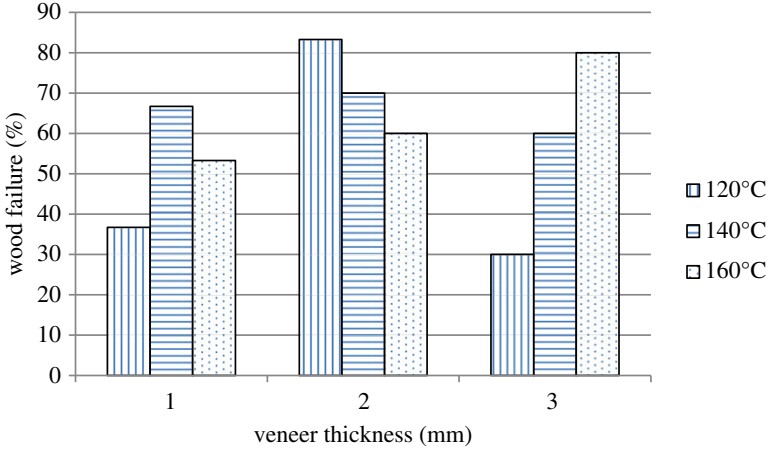

**Figure 2.** Per cent wood failure for rubberwood LVL made using 1, 2 and 3 mm veneer thicknesses at 120°C, 140°C and 160°C.

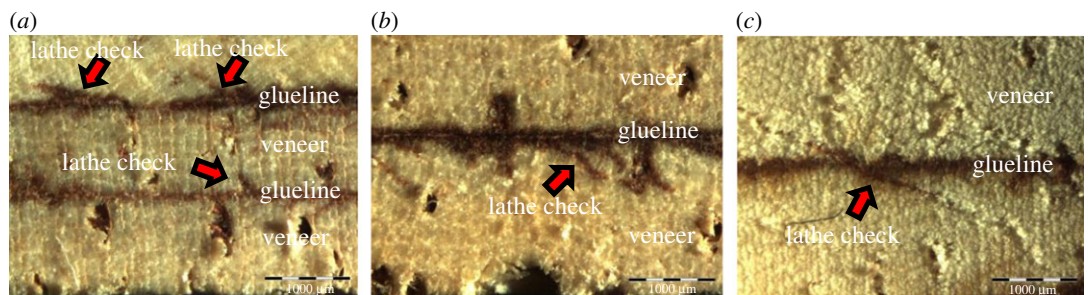

**Figure 3.** Glue penetration recorded by optical video microscope with ×75 magnification; rubberwood LVL produced by (*a*) 1 mm, (*b*) 2 mm and (*c*) 3 mm.

The gluebond shear strength result obtained from this research was in agreement with the studies by Rohumaa *et al*. [23] and DeVallance *et al*. [24], who reported that increasing the lathe check depth in thicker veneer significantly reduced the gluebond shear strength and the per cent wood failure of the panel. As shown in figure 3*c*, adhesive was unable to penetrate fully into the deep lathe checks on the surface of 3 mm veneer thickness to give stronger mechanical interlocking interaction and bonding quality [25,26]. Bonding quality was significantly affected by the pressing temperature as the adhesive penetration improved with the increase of pressing temperature [27]. At higher pressing temperature such as 160°C, the highest gluebond shear strength and the per cent wood failure were observed in LVL produced by 3 mm veneer thickness.

The gluebond shear strength decreases significantly as the veneer lathe check frequency increases [4]. In the previous published research, it was found that 1 and 2 mm rubberwood veneer has more lathe checks formation but generally with shallower lathe checks compared to the 3 mm veneer thickness [2]. The highest gluebond shear strength and the per cent wood failure were observed in LVL produced with 2 mm veneer thickness at 120°C, which has a higher lathe check frequency compared to 3 mm veneer. However, the depth of the lathe checks should be taken into consideration in this research. The presence of the higher lathe check frequency on 2 mm veneer loose side increased wettability, thus facilitating the optimum penetration of adhesive for stronger bonding (figure 3*b*). For LVL produced with 1 mm veneer thickness, the adhesive penetration was limited on the veneer surface only. In figure 3*a*, adhesive was distributed uniformly along the glueline between the surfaces of the veneers. The adhesive penetration in 1 mm veneer thickness was not as deep as in 2 mm veneer thickness.

# 4. Conclusion

Development of this LVL production will definitely increase the usability of these small-diameter rubberwood logs, which are currently only used as a low-grade woody material in particleboard or

fibreboard production. The effect of veneer thickness and pressing temperature interaction was highly significant on the physical properties and gluebond shear strength of three plies of rubberwood LVL. LVLs produced with thin veneer has the highest specific gravity but lowest dimensional stability. In terms of water absorption, LVL produced by 2 mm veneer thickness at 120°C has shown the lowest percentage after 2 h and 24 h immersion. LVLs produced with 2 mm veneer thickness at 120°C pressing temperature has the highest gluebond shear strength and the per cent wood failure. Veneer thickness of 2 mm was selected as the most suitable thickness for the LVL production. Pressing temperature at 120°C was sufficient to completely cure the glueline.

Ethics. We declare that the work submitted for the publication is original, has not been published elsewhere, accepted for publication elsewhere or under editorial review for publication elsewhere; and that all the authors mutually agree with its content and have approved the paper for release and submission. All the authors have declared no conflict of interest. The manuscript does not contain experiments using animals. At the same time, the manuscript does not contain human studies.

Data accessibility. Specific gravity, water absorption, tangential swelling, longitudinal swelling, radial swelling, volumetric swelling, gluebond shear strength and percentage wood failure data file has been uploaded to the Dryad Digital Repository: https://doi.org/10.5061/dryad.54s0mn2 [28].

Authors' contributions. P.S.K. carried out data collection, participated in analysis and interpretation of data and drafted the article; K.L.C. edited and revised the article critically for important intellectual content; P.S.H. agreed to be accountable for all aspects of the work in ensuring that questions related to the accuracy or integrity of any part of the work are appropriately investigated and resolved; E.S.B. edited and revised the article critically for important intellectual content; C.L.L. participated in analysis and interpretation of data; W.Z.G. participated in analysis and interpretation of data; R.D. carried out data collection. All authors gave final approval for publication.

Competing interests. We have no competing interests.

Funding. The authors are grateful for the financial support from co-author P.S.H. under the Higher Institution Centre of Excellence (HICoE) (grant no. 6369110) project at the Institute of Tropical Forestry and Forest Products which was given by the Ministry of Higher Education Malaysia (MOHE).

Acknowledgements. The authors are grateful for the financial support given by the Ministry of Higher Education Malaysia (MOHE) under the Higher Institution Centre of Excellence (HICoE) project at the Institute of Tropical Forestry and Forest Products.

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
