## [Reviewer comments · Royal Society Open Science]

Review History

RSOS-190879.R0 (Original submission)

Review form: Reviewer 1

Is the manuscript scientifically sound in its present form?

Yes

Are the interpretations and conclusions justified by the results?

Yes

Is the language acceptable?

Yes

Do you have any ethical concerns with this paper?

No

Have you any concerns about statistical analyses in this paper?

Yes

Recommendation?

Major revision is needed (please make suggestions in comments)

Comments to the Author(s)

Most of my comments are in the pdf file attached (Appendix A).

In general, this paper over-discuss the results. In my opinion, the authors should be more careful when discussing the results. There is sometimes a misuse of the Tukey's test: some results and discussion say the opposite of what give this statistical test.

Review form: Reviewer 2

Is the manuscript scientifically sound in its present form?

Yes

Are the interpretations and conclusions justified by the results?

Yes

Is the language acceptable?

Yes

Do you have any ethical concerns with this paper?

No

Have you any concerns about statistical analyses in this paper?

No

Recommendation?

Accept with minor revision (please list in comments)

Comments to the Author(s)

The paper can be interesting for the readers and brings new information on the use of rubberwood in wood technology. The analyzed process parameters are important in future development of the technology.

However, some drawbacks occur. These are:

1. The phrase "rubber log" is misleading and not precise. I suggest use "rubber wood log" throughout the text (title, abstract and maintext).
2. Two styles of reference citations occur in the text. Both numbers in square parentheses and name and year (p. 3 lines 5, 24; p. 6. line 56, 58; p. 7 lines 32, 35; 44).
3. Table 1 headings: "frequency" - are there any units? and "contact angle": I suggest use "contact angle after 10 seconds" or "contact angle in 10th second" ..

When these edits are done, I suggest accept.

Decision letter (RSOS-190879.R0)

31-Jul-2019

Dear Miss KHOO,

The editors assigned to your paper ("Physical Properties and Bonding Quality of LVL Produced with Veneers Peeled from Small Diameter Rubber Log") have now received comments from reviewers. We would like you to revise your paper in accordance with the referee and Associate Editor suggestions which can be found below (not including confidential reports to the Editor). Please note this decision does not guarantee eventual acceptance.

Please submit a copy of your revised paper before 23-Aug-2019. Please note that the revision deadline will expire at 00.00am on this date. If we do not hear from you within this time then it will be assumed that the paper has been withdrawn. In exceptional circumstances, extensions may be possible if agreed with the Editorial Office in advance. We do not allow multiple rounds of revision so we urge you to make every effort to fully address all of the comments at this stage. If deemed necessary by the Editors, your manuscript will be sent back to one or more of the original reviewers for assessment. If the original reviewers are not available, we may invite new reviewers.

- Data accessibility

If you wish to submit your supporting data or code to Dryad (<http://datadryad.org/>), or modify your current submission to dryad, please use the following link:
<http://datadryad.org/submit?journalID=RSOS&manu=RSOS-190879>

- **Competing interests**

- **Authors' contributions**

- **Acknowledgements**

- **Funding statement**

Kind regards,

Alice Power

Editorial Coordinator

on behalf of Dr Maria Charalambides (Associate Editor) and R. Kerry Rowe (Subject Editor)
openscience@royalsociety.org

Comments to Author:

Reviewers' Comments to Author:

Reviewer: 1

Comments to the Author(s)

Most of my comments are in the pdf file attached

In general, this paper over-discuss the results. In my opinion, the authors should be more carefull when discussing the results. There is sometimes a misusage of the Tukey's test: some results and discussion say the opposite of what give this statistical test.

Reviewer: 2

Comments to the Author(s)

The paper can be interesting for the readers and brings new information on the use of rubber wood in wood technology. The analyzed process parameters are important in future development of the technology.

However, some drawbacks occur. These are:

1. The phrase "rubber log" is misleading and not precise. I suggest use "rubber wood log" throughout the text (title, abstract and main text).
2. Two styles of reference citations occur in the text. Both numbers in square parentheses and name and year (p. 3 lines 5, 24; p. 6. line 56, 58; p. 7 lines 32, 35; 44.
3. Table 1 headings: "frequency" - are there any units? and "contact angle": I suggest use "contact angle after 10 seconds" or "contact angle in 10th second"..

When these edits are done, I suggest accept.

Author's Response to Decision Letter for (RSOS-190879.R0)

See Appendix B.

RSOS-190879.R1 (Revision)

Review form: Reviewer 1

Is the manuscript scientifically sound in its present form?

Yes

Are the interpretations and conclusions justified by the results?

No

Is the language acceptable?

Yes

Do you have any ethical concerns with this paper?

No

Have you any concerns about statistical analyses in this paper?

Yes

Recommendation?

Reject

Comments to the Author(s)

This manuscript is perfectly scientifically sound in its form, however, I strongly believe that most of the results cannot be trusted because they relies on 3 samples only. A statistical study may have been held correctly, but, in my experience of wood material and on the behalf of what we can fin in the litterature for experiments on wood, it is more like 30 samples that should be used rather than 3.

However, depending on the editor policy this paper may be published as it is written and the reader know it relies on 3 samples.

Review form: Reviewer 2**Is the manuscript scientifically sound in its present form?**

Yes

Are the interpretations and conclusions justified by the results?

Yes

Is the language acceptable?

Yes

Do you have any ethical concerns with this paper?

No

Have you any concerns about statistical analyses in this paper?

No

Recommendation?

Accept as is

Comments to the Author(s)

As all the suggested corrections have been done, the manuscript is ready for publication, I suggest accept.

Decision letter (RSOS-190879.R1)

24-Sep-2019

Dear Miss KHOO:

Manuscript ID RSOS-190879.R1 entitled "Physical Properties and Bonding Quality of LVL Produced with Veneers Peeled from Small Diameter Rubberwood Log" which you submitted to

Royal Society Open Science, has been reviewed. The comments from reviewer(s) are included at the bottom of this letter.

In view of the criticisms of the reviewer(s), I must decline the manuscript for publication in Royal Society Open Science at this time. However, a new manuscript may be submitted which takes into consideration these comments.

Please note that resubmitting your manuscript does not guarantee eventual acceptance, and that your resubmission will be subject to re-review by the reviewer(s) before a decision is rendered.

You will be unable to make your revisions on the originally submitted version of your manuscript. Instead, revise your manuscript using a word processing program and save it on your computer.

You may also click the below link to start the resubmission process (or continue the process if you have already started your resubmission) for your manuscript. If you use the below link you will not be required to login to ScholarOne Manuscripts.

*** PLEASE NOTE: This is a two-step process. After clicking on the link, you will be directed to a webpage to confirm. ***

https://mc.manuscriptcentral.com/rsos?URL_MASK=c715b0ed7c724507bce101d1c864ce01

Because we are trying to facilitate timely publication of manuscripts submitted to Royal Society Open Science, your resubmitted manuscript should be submitted by 23-Mar-2020. If you are unable to submit by this date please contact the Editorial Office for options.

I look forward to a resubmission.

on behalf of Dr Maria Charalambides (Associate Editor) and R. Kerry Rowe (Subject Editor)
openscience@royalsociety.org

Reviewer comments to Author:
Reviewer: 2

Comments to the Author(s)
As all the suggested corrections have been done, the manuscript is ready for publication, I suggest accept.

Reviewer: 1

Comments to the Author(s)

This manuscript is perfectly scientifically sound in its form, however, I strongly believe that most of the results cannot be trusted because they relies on 3 samples only. A statistical study may have been held correctly, but, in my experience of wood material and on the behalf of what we can fin in the litterature for experiments on wood, it is more like 30 samples that should be used rather than 3.

However, depending on the editor policy this paper may be published as it is written and the reader know it relies on 3 samples.

Author's Response to Decision Letter for (RSOS-190879.R1)

See Appendix C.

RSOS-191763.R0

Review form: Reviewer 1

Is the manuscript scientifically sound in its present form?

Yes

Are the interpretations and conclusions justified by the results?

Yes

Is the language acceptable?

Yes

Do you have any ethical concerns with this paper?

No

Have you any concerns about statistical analyses in this paper?

No

Recommendation?

Accept as is

Comments to the Author(s)

The authors made a very detailed reply showing their mastery of the subject. The article in it's current form explain to the reader what is done and thus I suggest to accept is as it is. However, I was not able to have access to the Dryad repository with the data of the experiments, the authors should fix it.

Decision letter (RSOS-191763.R0)

06-Nov-2019

Dear Miss KHOO,

I am pleased to inform you that your manuscript entitled "Physical Properties and Bonding Quality of LVL Produced with Veneers Peeled from Small Diameter Rubberwood Log" is now accepted for publication in Royal Society Open Science.

You have the opportunity to archive your accepted, unbranded manuscript, but access to the full text must be embargoed until publication.

Articles are normally press released. For this to be effective we set an embargo on news coverage corresponding to the publication date of the article. We request that news media and the authors do not publish stories ahead of this embargo (when final version of the article is available).

on behalf of Dr Maria Charalambides (Associate Editor) and R. Kerry Rowe (Subject Editor)
openscience@royalsociety.org

Associate Editor Comments to Author (Dr Maria Charalambides):

Please act on the reviewer's comment:

"However, I was not able to have access to the Dryad repository with the data of the experiments, the authors should fix it. "

Reviewer comments to Author:

Reviewer: 1

Comments to the Author(s)

The authors made a very detailed reply showing their mastery of the subject. The article in its current form explain to the reader what is done and thus I suggest to accept is as it is. However, I was not able to have access to the Dryad repository with the data of the experiments, the authors should fix it.

Appendix A**ROYAL SOCIETY
OPEN SCIENCE****Physical Properties and Bonding Quality of LVL Produced
with Veneers Peeled from Small Diameter Rubber Log**

Journal:	Royal Society Open Science
Manuscript ID	RSOS-190879
Article Type:	Research
Date Submitted by the Author:	10-Jun-2019
Complete List of Authors:	KHOO, PUI SAN; University Putra Malaysia, Institute of Tropical Forestry and Forest Products CHIN, KIT LING; University Putra Malaysia, Institute of Tropical Forestry and Forest Products H'NG, PAIK SAN; University Putra Malaysia, Institute of Tropical Forestry and Forest Products; Universiti Putra Malaysia Fakulti Perhutanan Bakar, Edi Suhaimi; Universiti Putra Malaysia, Fakulti Perhutanan LEE, CHUAN LI; University Putra Malaysia, Institute of Tropical Forestry and Forest Products GO, WEN ZE; Universiti Putra Malaysia Fakulti Perhutanan, Faculty of Forestry Dahali, Rasdianah; Universiti Putra Malaysia, Fakulti Perhutanan
Subject:	Materials science < CHEMISTRY, Materials science < ENGINEERING AND TECHNOLOGY
Keywords:	spindleless lathe technology, veneer thickness, pressing temperature, gluebond shear strength, rubberwood, small diameter log
Subject Category:	Engineering

Author-supplied statements

Relevant information will appear here if provided.

Ethics

Does your article include research that required ethical approval or permits?:

This article does not present research with ethical considerations

Statement (if applicable):

CUST_IF_YES_ETHICS :No data available.

Data

It is a condition of publication that data, code and materials supporting your paper are made publicly available. Does your paper present new data?:

Yes

Statement (if applicable):

Specific gravity, water absorption, tangential swelling, longitudinal swelling, radial swelling, volumetric swelling, gluebond shear strength and percentage wood failure data file has been uploaded to the dryad digital repository at: <https://doi.org/10.5061/dryad.54s0mn2>

Dryad review URL: <https://datadryad.org/review?doi=doi:10.5061/dryad.54s0mn2>

Conflict of interest

I/We declare we have no competing interests

Statement (if applicable):

CUST_STATE_CONFLICT :No data available.

Authors' contributions

This paper has multiple authors and our individual contributions were as below

Statement (if applicable):

- i. KHOO, PUI SAN- data collection, analysis and interpretation of data, drafting the article
- ii. CHIN, KIT LING- editing and revising the article critically for important intellectual content
- iii. H'NG, PAIK SAN- agreement to be accountable for all aspects of the work in ensuring that questions related to the accuracy or integrity of any part of the work are appropriately investigated and resolved
- iv. BAKAR, EDI SUHAIMI- editing and revising the article critically for important intellectual content
- v. LEE, CHUAN LI- analysis and interpretation of data
- vi. GO, WEN ZE- final approval of the version to be published
- vii. DAHALI, RASDIANA- acquisition of data

Physical Properties and Bonding Quality of LVL Produced with Veneers Peeled from Small Diameter Rubber Log

P.S. Khoo¹, K.L. Chin^{1*}, P.S. H'ng^{12*}, E.S. Bakar², C.L. Lee¹, W.Z. Go², R. Dahali²

¹*Institute of Tropical Forestry and Forest Products, Universiti Putra Malaysia, 43400 UPM Serdang, Selangor, Malaysia*

²*Faculty of Forestry, Universiti Putra Malaysia, 43400 UPM Serdang, Selangor, Malaysia*

Keywords: Rubberwood; Small diameter log; Spindleless lathe technology; Veneer thickness; Pressing temperature; Gluebond shear strength

1. Summary

The peeling of small diameter rubber logs from the current short-rotation practices, undoubtedly will produce lower-grade veneers compared to the veneers from the conventional planting rotation. Hence, this raises the question of the properties of the produced laminated veneer lumber (LVL) from veneers peeled from small diameter rubber logs using spindleless lathe technology. Different thicknesses of rubberwood veneers were peeled from rubber logs with diameter less than 20 cm using spindleless lathe. Three layer LVLs were prepared using phenol formaldehyde (PF) adhesive and hot pressed at different temperatures. During the peeling of veneer, lathe checks as deep as 30 to 60% of the veneer thickness are formed. This study showed that deep lathe check of 3 mm rubberwood veneer significantly reduced the gluebond shear strength of PF bonded LVL. In addition, lathe check frequency was also shown to influence bond strengths. The presence of higher lathe check frequency on 2 mm veneer increased the wettability, thus, facilitated optimum penetration of adhesive for stronger bonding. These findings stress the importance of measuring and considering the lathe check depth and frequency during lamination process to get a better understanding of bonding quality in veneer-based products.

2. Introduction

Rubberwood is a well-known species and highly requested by wood industry particularly for biocomposite production [1]. However, there has been a short supply of rubber logs for veneer production due to the inadequate diameter of the logs for veneer peeling. Instead of the conventional planting rotation of 25 to 30 years, nowadays, rubber plantations in Malaysia are managed under short planting cycle and the trees will be likely felled when they are around 15 years old due to the high demand of latex and rubberwood [2] which resulted with small diameter logs are harvested. The invention of spindleless lathe in recent years had encouraged the utilization of fast growing plantation species with small diameter logs in the production of structural composite lumber such as laminated veneer lumber (LVL) [3].

The mechanical properties of LVL generally depend upon a variety of factors particularly veneer quality (moisture content, density, lathe checks, surface roughness, thickness variation), adhesive quality (type of adhesive, mixture of resin, and viscosity) and bonding quality (wettability, glue spread rate, pressure, time and temperature) [4,5]. Numerous factors contribute to the veneer quality. The most common factors are due to log's characteristics (species, log diameter, log quality, specific gravity, wood pores, grain structure, juvenile and mature wood) [4].

*H'ng Paik San; Chin Kit Ling (ngpaiksan@gmail.com; kitling.chin419@gmail.com).

† Institute of Tropical Forestry and Forest Products, Universiti Putra Malaysia, 43400 UPM Serdang, Selangor, Malaysia

<https://mc.manuscriptcentral.com/rsos>

Owing to younger age and shorter planting cycle, juvenile wood is proportionally higher than mature wood [6]. This higher proportion of juvenile wood can have significant effect on the log quality, as well as veneer quality [6,7]. The presence of huge proportion of juvenile wood in a log will caused excessive shrinking and swelling, problems of warping, fuzzy grain, and general instability in the manufacture and use of the wood. These problems may occur in wood after sawing, veneering, drying and machining [8]. Darmawan et al. (2015) revealed that veneer with severe lathe checks will be produced when fast growing species with higher proportion of juvenile woods are peeled. Peeling veneers from small diameter log produced deeper lathe checks, longer lathe checks and larger interval between lathe checks [2,9,10]. Veneer produced from small diameter rubber log using spindleless lathes undoubtedly have very different properties compared to veneers produced from large diameter logs using conventional spindled lathes.

To date, however, there has been no published report on the production of LVL with veneers peeled from small diameter rubber logs using spindleless lathe. Due to the unique properties of veneer peeled from small diameter rubber log, the optimum parameters required for the production of LVL have to be determined. Therefore, the purpose of this research was to evaluate the physical properties and gluebond shear strength of rubberwood LVL manufactured from different veneer thicknesses at different pressing temperature. The LVL properties are important in determining the utilizing potential of LVL produced with veneers peeled from small diameter rubber log using spindleless lathe technology.

3. Materials and Methods

Small diameter rubber logs (between 15 and 18 cm) were peeled according to the method by Khoo et al. (2018), to produce veneer thickness with 1, 2 and 3 mm and the veneer properties were illustrated in Table 1;

Three layer LVL specimens were prepared from 1, 2 and 3 mm thickness rotary peeled rubberwood veneers with moisture content of $8\pm 2\%$ [11]. Phenol formaldehyde (PF) adhesive with 45% solid content were obtained from Aica Chemicals (M) Sdn. Bhd. Properties of the PF adhesive were as follows: specific gravity of 1.232 at 30°C; pH of 12.90 at 30°C; viscosity of 69 Cps at 30°C and gel time of 21 minutes at 105°C. Commercial filler was used with the PF adhesive. Double glueline spread rate of 200 g/m² was applied on the veneer surface, the veneer sheets were pressed together parallel to each other and the loose side of the veneer was placed towards the center of the boards. The three plies LVLs were hot pressed at 120, 140 and 160°C for 5 minutes with 7 kgf/cm² specific pressing pressure. After hot pressing, the LVLs were conditioned at temperature of $20\pm 3^\circ\text{C}$ and relative humidity of $65\pm 1\%$ until they reached the equilibrium moisture content of $10\pm 2\%$.

3.1 Evaluation

3.1.1 Moisture Content and Specific Gravity

Moisture content of the produced LVL was determined using conventional drying method according to ASTM D 4442-03. The specimens were oven-dried at $103\pm 2^\circ\text{C}$ until constant weight in order to determine the oven-dry weight. Moisture content of the specimens was calculated as follow:

$$\text{Moisture Content}(\%) = \frac{\text{Initial Weight (g)} - \text{Oven - dry Weight (g)}}{\text{Oven - dry Weight (g)}} \times 100\%$$

Specific gravity of test samples with dimension of 50 mm by 50 mm were determined according to ASTM D 2395-02. Oven-dry density and specific gravity of the specimens were calculated using formula as follow:

$$\text{Oven - dry density} \left(\frac{\text{g}}{\text{cm}^3} \right) = \frac{\text{Oven - dry mass}}{\text{Oven - dry volume}}$$

$$\text{Specific gravity} = \frac{\text{Oven - dry density}}{\text{Density of water}}$$

3.1.2 Water Absorption and Volumetric Swelling

Test specimens with dimension of 50 mm by 50 mm were weighed and the radial (thickness), tangential and longitudinal directions were measured before submerged in 25 mm of distilled water maintained at temperature of $20\pm 1^\circ\text{C}$. After a 2 hours submersion, the water was being removed and the specimens were suspend to drain for 10 ± 2 minutes in order to remove excess surface water. The specimens were weighed and the radial, tangential and longitudinal of the specimens was measured immediately. After that, the specimens were submerged for an additional 22 hours and followed by the weighing and measuring procedure mentioned above. After submersion, the specimens were put in oven at $103\pm 2^\circ\text{C}$ to calculate the moisture content based on oven-dry weight. Based on ASTM D 1037-12, the percentage of water absorption, radial, tangential, longitudinal and volumetric swelling were determined using formula as follow:

$$\text{Water Absorption (\%)} = \frac{\text{Final Weight (g)} - \text{Initial Weight (g)}}{\text{Initial Weight (g)}} \times 100\%$$

$$\text{Swelling (\%)} = \frac{\text{Final Length (mm)} - \text{Initial Length (mm)}}{\text{Initial Length (mm)}} \times 100\%$$

$$\text{Volumetric Swelling (\%)} = \frac{\text{Final Volume (cm}^3\text{)} - \text{Initial Volume (cm}^3\text{)}}{\text{Initial Volume (cm}^3\text{)}}$$

3.1.3 Gluebond Shear Strength

For gluebond shear strength, the cutting pattern for testing specimens  as shown in Fig. 1.

Testing specimens were tested according to the ASTM D 906 using INSTRON Universal Testing Machine. Load was applied continuously throughout the test at a uniform rate of motion of the movable crosshead of the testing machine of 4 mm/min. The shear strength of each specimen was calculated from the following formula:

$$\text{Shear strength} = \frac{F}{l \times b}$$

Where,

F = failing force of the specimen, in Newtons;

l = length of the shear area, in mm;

b = width of the shear area, in mm.

The glue penetration of the rubberwood veneer was examined using an Olympus SZX12 stereo microscope to evaluate the interaction between the adhesive and rubberwood veneers. Specimens were taken at the cross section of each LVL panel. The LVL specimens of size 10 mm wide x 10 mm long were cut for assessment. Specimen was examined under microscope at $75\times$ magnification.

4. Results and Discussion

As shown in Table 2, the interaction between veneer thickness and pressing temperature was analyzed using one-way analysis of variance (ANOVA). The interaction of veneer thickness and pressing temperature has highly significant effect on the physical properties and gluebond shear strength of 3-ply rubberwood LVLs. The significant mean values were compared using Tukey's test and the results were tabulated in Table 2.

4.1 Moisture Content and Specific Gravity

After conditioning for two weeks, the equilibrium moisture content of LVLs produced from different thickness of rubberwood veneers were $13\pm 2\%$. Based on the verdicts there was an increase in the density of rubberwood LVLs compared to solid rubberwood. From previous research, air dry density of solid rubberwood generally ranged from 650 to 706 kg/m³, stated by [2]. In this study, it shows the air dry density

of rubberwood LVLs ranged from 736 to 857 kg/m³ was higher than the density of solid rubberwood. The increase in density of 3-ply rubberwood laminated veneer lumber is probably due to addition of adhesive. The density of phenol formaldehyde resin used in this study was 1.2 g/cm³, which is much higher than that of the substrate. Hence, the total mass of the panels was greatly affected by the density of the adhesive [12].

One way analysis of variance (Table 2) clearly shows that the specific gravity of rubberwood LVLs are highly significantly influenced by the interaction between veneer thickness and pressing temperature. Highest specific gravity can be found in LVL produced with 1 mm veneer thickness at 120°C whereas the lowest specific gravity rubberwood LVLs were produced using 3 mm veneer thickness at 100°C. With increasing pressing temperature, reduction in specific gravity was more drastically in rubberwood LVL produced with 1 mm veneer thickness compared to rubberwood LVL produced with 2 and 3 mm veneer thickness. Reduction in specific gravity is highly related to the degradation and changes of both chemical and physical properties due to the exposure of wood to higher temperature [6]. Between 100 to 200°C, dehydration and decarboxylation processes happen and wood generate water vapour, carbon dioxide and volatile organic compounds (VOC); thus the carbohydrate polymers in wood depolymerize and degrade slowly as the temperature increase [13].

Regardless of pressing temperature, LVL produced with 3 mm veneer thickness has significantly lower specific gravity compared to 1 and 2 mm veneer thickness. This variation is mainly attributed to the presence of more pores and void volume in thick veneer, resulted in lower specific gravity. High specific gravity in LVL produced with thin veneer generally due to the presence of less pores and void volume [14].

4.2 Water Absorption and Volumetric Swelling

The interaction between veneer thickness and pressing temperature has highly significant effect on the percentage of water absorption, radial swelling, longitudinal swelling, tangential and volumetric swelling after two and 24 hours water immersion (Table 2). Within the first two hours immersion, rubberwood LVL produced from 1 mm veneer thickness at 160°C has the highest percentage of water absorption and radial swelling due to the permeability of thin veneer which allow for efficient water transport compared to thicker veneer [15]. As water molecule forming hydrogen bonds with active hydroxyl groups in cell wall polymers, the cell wall expands to accommodate the water molecules [16]. This results in higher percentage of water absorption and radial swelling in thin veneer compared to thick veneer. At the same time, pressing LVL at higher temperature might accelerated the condensation process of glucose and more bubbles will be formed within the glucose. The void created by the bubbles might provide a void volume for water molecule to fill up; hence the percentage of water absorption and radial swelling increased with increasing pressing temperature [13,17].

After 24 hours immersion, rubberwood LVL produced from 3 mm veneer thickness at 140°C has the highest value in water absorption. This might be due to more porosity in 3 mm veneer thickness compared to 1 and 2 mm veneer thickness. This result was supported by the specific gravity result, which mentioned that the rubberwood LVL produced from 3 mm veneer thickness at 140°C has lowest specific gravity. Since the volume of the lumina increase as decreasing specific gravity, the void volume in cell lumina will be replaced by free water until the maximum moisture content is achieved [18]. This resulted in the highest percentage of water absorption after the sample was immersed for 24 hours. On the other hand, lowest percentage of water absorption after 24 hours immersion was significantly obtained by LVL produced from 1 and 2 mm veneer thickness at 120°C; which are 46.46% and 45.98%, respectively [19]. This result was in agreement with specific gravity result, which shown that as increasing specific gravity, the volume of the lumina decrease. This decreases the maximum moisture content because less void is available for free water [18].

Based on Table 2, swelling in longitudinal direction was 0.4±0.2% and 1.1±0.5% after two and 24 hours immersion, respectively. In fact, there is no longitudinal swelling occurs in most species stated by Uysal (2005). However, rubberwood is expected to have higher longitudinal swelling compared to other species due to presence of tension wood [21]. Previous research from Sik et al. (2009) reported that longitudinal shrinkage in rubberwood contains tension wood up to 0.67% compared to normal shrinkage which is about 0.1-0.3%. In this study, the longitudinal swelling of rubberwood LVL after two hours immersion (0.4±0.2%) is considerably lower compared to the longitudinal shrinkage in rubberwood contains tension wood (0.67%). Rubberwood

LVL produced by thin veneer has a relatively lower longitudinal swelling after 24 hours immersion compared to thicker veneer. By converting rubberwood into LVL, tension wood will disperse more uniformly in thin veneer and it may cause a minor impact to the longitudinal swelling. With the shorter planting cycle, harvested rubber logs are certainly smaller in diameter and lower in wood quality due to the presence of higher proportion of juvenile wood [2,6,7]. Juvenile wood has larger microfibril angle in fibres results in greater longitudinal shrinkage and lesser tangential shrinkage [6,7].

LVL produced by 3 mm veneer thickness was found to be the most dimensional stable panels with the least radial swelling, tangential swelling and volumetric swelling after two and 24 hours. Nevertheless, LVL with 3 mm veneer thickness has significant increment in the percentage of water absorption from two hours immersion to 24 hours immersion. This increment due to the LVL had reached the fiber saturation point within first two hours immersion. As hydroxyl groups in cell wall polymers have been saturated with water molecules, the cell wall no longer expands to accommodate the water molecules [16]. Wood is dimensionally unstable within the first two hours immersion due to the moisture content below fiber saturation point (usually 30% moisture content). Once the moisture content is beyond the fiber saturation point, the excess free water will only fills lumens and makes wood heavier, but does not contribute to further expansion [20].

4.3 Gluebond Shear Strength

In terms of gluebond shear strength, the effect of veneer thickness and pressing temperature interaction was highly significant. Not only the information regarding to the gluebond shear strength is crucial, but percent wood failure is also important for the evaluation of bonding quality. The LVL failures after gluebond shear test were observed and evaluated visually. The result was shown in Fig. 2.

According to Table 2 and Fig. 2, the highest gluebond shear strength and percent wood failure were observed in LVL produced with 2 mm veneer thickness at 120°C whereas the lowest gluebond shear strength and percent wood failure were obtained by LVL produced with 3 mm veneer thickness at 120°C. Previous research by Khoo et al. (2018) revealed that thicker veneer usually contained deeper lathe check depth, longer lathe check length but lower lathe check frequency. The gluebond shear strength result obtained from this research was in agreement with Rohumaa et al. (2013) and DeVallance et al. (2007), who reported that increasing lathe check depth in thicker veneer significantly reduce the gluebond shear strength and the percent wood failure of the panel. As shown in Fig. 3(c), adhesive was unable to penetrate fully into the deep lathe checks on the surface of 3 mm veneer thickness in order to give stronger mechanical interlocking interaction and bonding quality [25,26]. Bonding quality was significantly affected by the pressing temperature as the adhesive penetration improved with the increasing of pressing temperature. At higher pressing temperature such as 160°C, the highest gluebond shear strength and percent wood failure were observed in LVL produced by 3 mm veneer thickness [1,17,27].

Darmawan et al. (2015) stated that the gluebond shear strength decrease significantly as the veneer lathe check frequency increases. In previous published research, it was found that 1 and 2 mm rubberwood veneer has more lathe checks formation but generally with shallower lathe checks compared to 3 mm veneer thickness [2]. Highest gluebond shear strength and percent wood failure were observed in LVL produced with 2 mm veneer thickness at 120°C which have a higher lathe check frequency compared to 3 mm veneer. However, the depthness of the lathe checks should be taken into consideration in this research. The presence of higher lathe check frequency on 2 mm veneer loose side increased wettability, thus, facilitated optimum penetration of adhesive for stronger bonding (Fig. 3(b)). For LVL produced with 1 mm veneer thickness, the adhesive penetration was limited on the veneer surface only. In Fig. 3(a), adhesive was distributed uniformly along the glueline between the surfaces of the veneers. The adhesive penetration in 1 mm veneer thickness was not as deep as in 2 mm veneer thickness.

5. Conclusion

Development of this LVL production will definitely increase the usability of these small diameter rubber logs, which are currently only used as a low grade woody material in particleboard or fiberboard production. The effect of veneer thickness and pressing temperature interaction was highly significant on the physical

properties and gluebond shear strength of 3 plies rubberwood LVL. LVLs produced with thin veneer has the highest specific gravity but lowest dimensional stability. In terms of water absorption, LVL produced by 2 mm veneer thickness at 120°C has shown the lowest percentage after two and 24 hours immersion. LVLs produced with 2 mm veneer thickness at 120°C pressing temperature has the highest gluebond shear strength and percent wood failure. 2 mm veneer thickness was selected as the most suitable thickness for the LVL production. Pressing temperature at 120°C was sufficient in order to completely cured the glueline.

Acknowledgments

The authors are grateful for the financial support given by the Ministry of Higher Education Malaysia (MOHE) under the Higher Institution Centre of Excellence (HICoE) project at the Institute of Tropical Forestry and Forest Products.

Ethical Statement

I declare that the work submitted for the publication is original, has not been published elsewhere, accepted for publication elsewhere or under editorial review for publication elsewhere; and that all the authors mutually agree with its content and have approved the paper for release and submission. All the authors have declared no conflict of interest. The manuscript does not contain experiments using animals. In the same time, the manuscript does not contain human studies.

Funding Statement

The authors are grateful for the financial support from co-author H'ng Paik San under the Higher Institution Centre of Excellence (HICoE) project at the Institute of Tropical Forestry and Forest Products which given by the Ministry of Higher Education Malaysia (MOHE).

Data Accessibility

Specific gravity, water absorption, tangential swelling, longitudinal swelling, radial swelling, volumetric swelling, gluebond shear strength and percentage wood failure data file has been uploaded to the dryad digital repository at: <https://doi.org/10.5061/dryad.54s0mn2>

Dryad review URL: <https://datadryad.org/review?doi=doi:10.5061/dryad.54s0mn2>

Authors' Contributions

- i. KHOO, PUI SAN- data collection, analysis and interpretation of data, drafting the article
- ii. CHIN, KIT LING- editing and revising the article critically for important intellectual content
- iii. H'NG, PAIK SAN- agreement to be accountable for all aspects of the work in ensuring that questions related to the accuracy or integrity of any part of the work are appropriately investigated and resolved
- iv. BAKAR, EDI SUHAIMI- editing and revising the article critically for important intellectual content
- v. LEE, CHUAN LI- analysis and interpretation of data
- vi. GO, WEN ZE- final approval of the version to be published
- vii. DAHALI, RASDIANA- acquisition of data

Competing Interests

'We have no competing interests.'

References

1. Ratnasingam, J, Ioras, F, Wenming, L. 2011. Sustainability of the rubberwood sector in Malaysia. *Not. Bot. Horti Agrobot. Cluj-Napoca* 39, 305–311. (doi:10.15835/nbha3927195)
2. Khoo, PS, H'ng, PS, Chin, KL, Bakar, ES, Maminski, M, Raja-Ahmad, RN, Lee, CL, Ashikin, SN, Saharudin, MH. 2018. Peeling of small diameter rubber log using spindleless lathe technology: evaluation of veneer properties from outer to inner radial section of log at different veneer thicknesses. *Eur. J. Wood Wood Prod.* 76, 1335–1346. (doi:10.1007/s00107-018-1300-5)
3. Bal, BC. 2016. Some technological properties of laminated veneer lumber produced with fast-growing Poplar and Eucalyptus. *Maderas. Cienc. y Tecnol.* 18, 413–424. (doi:10.4067/S0718-221X2016005000037)
4. Darmawan, W, Nandika, D, Massijaya, Y, Kabe, A, Rahayu, I, Denaud, L, Ozarska, B. 2015. Lathe check characteristics of fast growing sengon veneers and their effect on LVL glue-bond and bending strength. *J. Mater. Process. Technol.* 215, 181–188. (doi:10.1016/j.jmatprotec.2014.08.015)
5. Dundar, T, Akbulut, T, Korkut, S. 2008. The effects of some manufacturing factors on surface roughness of sliced Makoré (Tieghemella heckelii Pierre Ex A.Chev.) and rotary-cut beech (*Fagus orientalis* L.) Veneers. *Build. Environ.* 43, 469–474. (doi:10.1016/j.buildenv.2007.01.002)
6. Hillis, WE. 1984. High temperature and chemical effects on wood stability Part 1. *Wood Sci. Technol.* 18, 281–293. (doi:10.1007/BF00354753)
7. Rahayu, I. 2016. Characteristics of Lathe Check and Surface Roughness of Fast Growing Wood Veneers and Their Performance on Laminated Veneer Lumber. Doctoral dissertation, ENSAM, Paris.
8. Maeglin, R. 1987. Juvenile wood, tension wood and growth stress effects on processing hardwoods. In: *Applying the Latest Research to Hardwood Problems*. Proceedings of the 15th Annual Hardwood Symposium of the Hardwood Research Council, pp 100–108.
9. Pałubicki, B, Marchal, R, Butaud, JC, Denaud, LE, Bléron, L, Collet, R, Kowaluk, G. 2010. A method of lathe checks measurement; SMOF device and its Software. *Eur. J. Wood Wood Prod.* 68, 151–159. (doi:10.1007/s00107-009-0360-y)
10. Denaud, LE, Bléron, L, Ratle, A, Marchal, R. 2007. Online control of wood peeling process: Acoustical and vibratory measurements of lathe checks frequency. *Ann. For. Sci.* 64, 569–575. (doi:10.1051/forest:2007034)
11. Bekhta, P, Ortyńska, G, Sedliaciak, J. 2014. Properties of Modified Phenol-Formaldehyde Adhesive for Plywood Panels Manufactured from High Moisture Content Veneer. *Drv. Ind. Znan. časopis za pitanja Drv. Tehnol.* 65, 293–301. (doi:10.5552/drind.2014.1350)
12. Sulaiman, O, Salim, N, Hashim, R, Yusof, LHM, Razak, W, Yunus, NYM, Hashim,

1 WS, Azmy, MH. 2009. Evaluation on the suitability of some adhesives for laminated veneer lumber from oil palm trunks. *Mater. Des.* 30, 3572–3580. (doi:10.1016/j.matdes.2009.02.027)

2 13. Diertenberger, MA, Hasburgh, LE. 2016. *Wood Products: Thermal Degradation and Fire*, Reference Module in Materials Science and Materials Engineering. Elsevier Ltd. (doi:10.1016/B978-0-12-803581-8.03338-5)

3 14. Vick, CB. 1999. *Adhesive Bonding of Wood Materials*, Wood Handbook: Wood as an Engineering Material. Madison, WI: USDA Forest Service, Forest Products Laboratory, General technical report FPL; GTR-113, pp 9.1-9.24. (doi:10.1016/0079-6425(76)90040-2)

4 15. Ramage, MH, Burrige, H, BusseWicher, M, Fereday, G, Reynolds, T, Shah, DU, Wu, G, Yu, L, Fleming, P, DensleyTingley, D, Allwood, J, Dupree, P, Linden, PF, Scherman, O. 2017. The wood from the trees: The use of timber in construction. *Renew. Sustain. Energy Rev.* 68, 333–359. (doi:10.1016/j.rser.2016.09.107)

5 16. Rowell, RM, Youngs, RL. 1981. Dimensional Stabilization of Wood in Use, Forest Products Lab Madison WI.

6 17. Sedliačik, J, Bekhta, P, Potapova, O. 2010. Technology of low-temperature production of plywood bonded with modified phenol-formaldehyde resin. *Wood Res.* 55, 123–130.

7 18. Glass, SV, Zelinka, SL. 2010. Moisture Relations and Physical Properties of Wood. In: *Wood Handbook: Wood as an Engineering Material*, pp 19. (doi:General Technical Report FPL-GTR-190)

8 19. Shukla, SR, Kamdem, DP. 2009. Properties of laboratory made yellow poplar (*Liriodendron tulipifera*) laminated veneer lumber: Effect of the adhesives. *Eur. J. Wood Wood Prod.* 67, 397–405. (doi:10.1007/s00107-009-0333-1)

9 20. Uysal, B. 2005. Bonding strength and dimensional stability of laminated veneer lumbers manufactured by using different adhesives after the steam test. *Int. J. Adhes. Adhes.* 25, 395–403. (doi:10.1016/j.ijadhadh.2004.11.005)

10 21. Ratnasingam, J, Grohmann, R, Scholz, F. 2010. Drying quality of rubberwood: An industrial perspective. *Eur. J. Wood Wood Prod.* 68, 115–116. (doi:10.1007/s00107-009-0353-x)

11 22. Sik, HS, Choo, KT, Sarani, Z, Sahrim, A, How, SS, Omar, MKM. 2009. Influence of Drying Temperature on the Physical and Mechanical Properties of Rubberwood. *J. Trop. For. Sci.* 21, 181–189.

12 23. Rohumaa, A, Hunt, CG, Hughes, M, Frihart, CR, Logren, J. 2013. The influence of lathe check depth and orientation on the bond quality of phenol-formaldehyde - Bonded birch plywood. *Holzforschung* 67, 779–786. (doi:10.1515/hf-2012-0161)

13 24. DeVallance, DB, Funck, JW, Reeb, JE. 2007. Douglas-fir plywood gluebond quality as influenced by veneer roughness, lathe checks, and annual ring characteristics. *J. For. Prod.* 57, 21-29.

14 25. Frihart, CR. 2005. *Wood Adhesion and Adhesives Chapter 9*. In: *Handbook of wood chemistry and wood composites*, CRC Press, New York.

15 26. Kurt, R, Cil, M. 2012. Effects of press pressures on glue line thickness and properties of laminated veneer lumber glued with phenol formaldehyde adhesive. *BioResources* 7, 5346–5354. (doi:10.15376/biores.7.3.4341-4349)

16 27. Kamke, F, Lee, JN. 2007. Adhesive Penetration in Wood — a Review. *Wood Fiber Sci.* 39, 205–220.

Tables

Table 1 Properties of rotary peeled rubberwood veneers obtained by using spindleless lathe (Khoo et al. 2018)

Veneer Thickness (mm)	Lathe Check Properties		Contact Angle (°C) at 10 Seconds
	Depth (%)	Frequency	
1	30±10	35±10	15
2	50±15	30±10	8
3	60±10	25±5	34

Table 2 Mean values of interactions between veneer thickness and pressing temperature on the physical properties and gluebond shear strength of rubberwood LVLs

Thickness (mm)	Temperature (°C)	Properties					
		SG	WA2 (%)	WA24 (%)	RS2 (%)	RS24 (%)	LS2 (%)
1	120	0.776 ^c (0.02)	29.28 ^{ab} (0.87)	46.46 ^a (1.44)	2.413 ^b (0.15)	5.297 ^d (0.17)	0.502 ^{dc} (0.02)
	140	0.710 ^{cd} (0.03)	34.66 ^{cd} (2.52)	51.14 ^{bc} (3.74)	3.109 ^c (0.28)	5.619 ^d (0.32)	0.259 ^a (0.02)
	160	0.674 ^{bc} (0.04)	35.82 ^d (2.90)	48.29 ^{ab} (3.61)	2.951 ^c (0.08)	5.654 ^d (0.33)	0.444 ^c (0.03)
2	120	0.737 ^{de} (0.02)	26.11 ^a (2.18)	45.98 ^a (2.33)	1.594 ^a (0.09)	3.648 ^{bc} (0.23)	0.483 ^{cde} (0.04)
	140	0.719 ^d (0.02)	29.62 ^{ab} (2.24)	47.08 ^{ab} (0.72)	2.113 ^b (0.20)	3.722 ^{bc} (0.26)	0.367 ^b (0.03)
	160	0.707 ^{cd} (0.03)	32.24 ^{bcd} (2.93)	47.49 ^{ab} (2.15)	2.155 ^b (0.15)	3.281 ^{ab} (0.20)	0.525 ^c (0.04)
3	120	0.664 ^{ab} (0.02)	32.49 ^{bcd} (3.19)	51.14 ^{bc} (3.86)	1.449 ^a (0.14)	3.035 ^a (0.17)	0.372 ^b (0.03)
	140	0.631 ^a (0.02)	34.86 ^{cd} (2.50)	54.02 ^c (2.53)	1.738 ^a (0.11)	3.126 ^a (0.06)	0.470 ^{cde} (0.02)
	160	0.667 ^{ab} (0.02)	31.33 ^{bc} (2.92)	49.58 ^{abc} (1.73)	2.392 ^b (0.11)	3.991 ^c (0.19)	0.463 ^{cd} (0.03)
Pr > F		**	**	**	**	**	**

Thick-ness (mm)	Tempe-rature (°C)	Properties					
		LS24 (%)	TS2 (%)	TS24 (%)	VS2 (%)	VS24 (%)	GBS (MPa)
1	120	0.863 ^b (0.08)	2.269 ^b (0.21)	4.212 ^{de} (0.40)	5.128 ^{cd} (0.21)	10.591 ^{de} (0.65)	3.10 ^{ab} (0.04)
	140	1.002 ^{bc} (0.09)	2.796 ^c (0.18)	4.266 ^{de} (0.15)	6.868 ^f (0.16)	11.442 ^e (0.32)	4.14 ^{cd} (0.06)
	160	0.602 ^a (0.04)	2.291 ^b (0.19)	3.599 ^b (0.02)	5.933 ^e (0.39)	10.329 ^{cd} (0.52)	3.56 ^{bc} (0.13)
2	120	1.261 ^{ef} (0.07)	2.200 ^{ab} (0.14)	4.588 ^e (0.15)	4.322 ^b (0.28)	9.472 ^{bc} (0.62)	5.91 ^f (0.35)
	140	1.552 ^g (0.14)	2.307 ^b (0.22)	4.115 ^{cd} (0.21)	5.029 ^{cd} (0.49)	8.293 ^a (0.47)	3.88 ^{cd} (0.29)
	160	0.626 ^a (0.01)	2.579 ^{bc} (0.16)	3.951 ^{bcd} (0.22)	5.174 ^d (0.07)	8.595 ^{ab} (0.30)	4.31 ^d (0.30)
3	120	1.196 ^{de} (0.01)	1.869 ^a (0.13)	3.722 ^{bc} (0.15)	3.601 ^a (0.27)	8.078 ^a (0.48)	2.58 ^a (0.18)
	140	1.410 ^{fg} (0.12)	2.558 ^{bc} (0.22)	3.873 ^{bcd} (0.18)	4.537 ^{bc} (0.35)	8.459 ^a (0.55)	3.74 ^{cd} (0.17)
	160	1.064 ^{cd} (0.10)	1.841 ^a (0.15)	3.010 ^a (0.15)	4.361 ^b (0.31)	8.358 ^a (0.20)	5.05 ^c (0.13)
Pr > F		**	**	**	**	**	**

Specific gravity (SG), water absorption after 2 hours immersion (WA2), water absorption after 24 hours immersion (WA24), radial swelling after 2 hours immersion (RS2), radial swelling after 24 hours immersion (RS24), longitudinal swelling after 2 hours immersion (LS2), longitudinal swelling after 24 hours immersion (LS24), tangential swelling after 2 hours immersion (TS2), tangential swelling after 24 hours immersion (TS24), volumetric swelling after 2 hours immersion (VS2), volumetric swelling after 24 hours immersion (VS24), gluebond shear strength (GBS)

n.s. no significant

* significant at $P < 0.05$

** significant at $P < 0.01$

Values in parentheses indicate standard deviation

Figure 1. Cutting pattern of gluebond shear test specimen

Figure 2. Percent wood failure for rubberwood LVL made using 1, 2 and 3 mm veneer thickness at 120, 140 and 160°C

Figure 3. Glue penetration recorded by optical video microscope with $\times 75$ magnification; rubberwood LVL produced by (a) 1 mm, (b) 2 mm and (c) 3 mm

Appendix B

RESPONSE TO REVIEWER COMMENTS

Physical Properties and Bonding Quality of LVL Produced with Veneers Peeled from Small Diameter Rubberwood Log

Thank you for the thoughtful reviews of our manuscript. We take concerns seriously and have addressed them to the best of our abilities. Changes have been made as suggested by the reviewers. Some of the more notable changes are listed as below;

- 1) Numerous (p. 3 lines 56) (Reviewer #1)
 - ✓ Corrected

- 2) There is no information about the number of samples. It should be given. (Reviewer #1)
 - ✓ Each treatment was carried out in three replicates.
 - ✓ Information about the number of samples had been added in the manuscript (Materials and Methods).

- 3) Use 'as' instead 'are' (p. 5 lines 27) (Reviewer #1)
 - ✓ Corrected

- 4) Use 'was' instead of 'is' (p. 6 lines 2) (Reviewer #1)
 - ✓ Corrected

- 5) There is no influence of the temperature on SG according to the Tukey's test: this whole paragraph seems wrong (p. 6 lines 9 - 11). (Reviewer #1)
 - ✓ Analysis of variance result (Table 2) has shown the effect of pressing temperature was highly significant on the specific gravity of LVL.
 - ✓ Based on Table 2, the effect of pressing temperature was highly significant on the specific gravity of LVL. With increasing pressing temperature, reduction in specific gravity was more prominent in rubberwood LVL produced with 1 mm veneer thickness.
 - ✓ This sentence had been added in the manuscript.

- 6) It is definitively not high for WA. Please consider revising the sentence (p. 6 lines 26 - 29). (Reviewer #1)
 - ✓ Within the first two hours immersion, rubberwood LVL pressed with higher temperature has significantly higher percentage of radial swelling compared to others.
 - ✓ This sentence has been revised to address the reviewer's concerns.

- 7) It is not statistically proved (p. 6 lines 32). (Reviewer #1)
 - ✓ Correction was made to address the reviewer's concerns.

8) I believe this whole explanation is too much for unclear results (p. 6 lines 36 - 37). (Reviewer #1)

- ✓ The entire explanation has been revised.
- ✓ Pressing LVL with high temperature softens the wood and lead to compression of wood [16]. With the increasing pressing temperature, the compression of wood increase [17]. However, these compression stresses build up during hot pressing are released (springback) when panels immersed in water [18] which resulted in higher radial swelling.

9) Remove "is" (p. 6 lines 57) (Reviewer #1)

- ✓ Corrected

10) Please give the units for frequency this number looks strange (Table 1). (Reviewer #1)

- ✓ "Frequency" represents the number of lathe checks present on the veneer loose side within a 5 cm length. It is unitless. "Frequency" was replaced with "Frequency per 5 cm".

11) Please add the number of sample for each line (Table 2). (Reviewer #1)

- ✓ Information about the number of samples was added in Materials and Methods.

12) Does this cut should not end at half the middle veneer thickness like the one on the right? (Figure 1). (Reviewer #1)

- ✓ Yes, this cut should end at half the middle like the one on the right.
- ✓ Correction was made to address the reviewer's concerns.

13) The phrase "rubber log" is misleading and not precise. I suggest use "rubber wood log" throughout the text (title, abstract and main text). (Reviewer #2)

- ✓ Correction was made to address the reviewer's concerns.

14) Two styles of reference citations occur in the text. Both numbers in square parentheses and name and year (p. 3 lines 5, 24; p. 6. line 56, 58; p. 7 lines 32, 35; 44. (Reviewer #2)

- ✓ Correction was made to address the reviewer's concerns.

15) Table 1 headings: "frequency" - are there any units? and "contact angle": I suggest use "contact angle after 10 seconds" or "contact angle in 10th second". (Reviewer #2)

- ✓ "Frequency" represents the number of lathe checks present on the veneer loose side within a 5 cm length. It is unitless. "Frequency" was replaced with "Frequency per 5 cm".

Appendix C

RESPONSE TO REVIEWER COMMENTS

RSOS

Physical Properties and Bonding Quality of LVL Produced with Veneers Peeled from Small Diameter Rubberwood Log

Thank you for the thoughtful reviews of our manuscript. We take concerns seriously and have addressed them to the best of our abilities.

Reviewer comments to Author:

1) As all the suggested corrections have been done, the manuscript is ready for publication, I suggest accept. (Reviewer #2).

✓ Thank you for your thoughtful and constructive comments about our paper.

2) This manuscript is perfectly scientifically sound in its form, however, I strongly believe that most of the results cannot be trusted because they rely on 3 samples only. A statistical study may have been held correctly, but, in my experience of wood material and on the behalf of what we can find in the literature for experiments on wood, it is more like 30 samples that should be used rather than 3. However, depending on the editor policy this paper may be published as it is written and the reader know it relies on 3 samples. (Reviewer #1)

✓ Treatment replicates (sample size) are used to measure variation in the experiment so that statistical tests can be applied to evaluate differences. There is a widespread belief that large samples are ideal for research or statistical analysis. However, this is not always true. Very small sample size undermines the internal and external validity of a study. While, very large sample size tends to transform small differences into statistically significant differences - even when they are insignificant. As a result, many researchers are misguided, which may lead to failure in treatment decisions.

✓ We would also like to clarify that we did not simply perform the number of treatment replicates just by random. The number of samples/replicates were calculated based on the results of previous experiments and our experience on laminated veneer lumber (which is a very uniform product). We truly believe that understanding the purpose of the experiment and the capability of the procedure provides the most accurate results at the lowest cost. It enables the scientist to best determine how to balance the number of treatment replicates against the cost. The use of sample size calculation directly influences research findings. We highly agree that in the absence of this calculation, the findings of the study should be interpreted with caution. Below is the calculation on how we estimate the number of treatment replicates needed (This calculation was not created by us. It is a well-known calculation among researchers);

- ✓ To determine the number of treatment replicates to use, the following formula should be used (**Formula 1**). Results on the replicates needed for each testing properties as shown in **Table 1**.

$$\#reps = 2 \left(Z_{\frac{\alpha}{2}} + Z_{\beta} \right) \left(\frac{\sigma}{\delta} \right)^2 \quad \text{----- (Formula 1)}$$

Where,

$Z_{\frac{\alpha}{2}}$ = Type I error, $\alpha = 0.05$. According to table, the $Z_{\frac{0.05}{2}}$ value will be 1.96.

Z_{β} = Type II error, 0.20. According to table, the $Z_{0.20}$ value will be 0.84.

δ = True difference to be detected

σ = Standard deviation obtained from previous experiments

Calculation of the number of treatment replicates depends on:

1. An estimate of Standard deviation obtained from previous experiments.
 2. The size of the difference (δ) to be detected.
 3. The assurance with which it is desired to detect the difference (i.e., Power of the test = $1-\beta$).
 4. The level of significance to be used in the actual experiment (i.e., Type I error).
 5. The test required, whether a one-tail or two-tail test.
- ✓ To ensure the repeatability and reliability of the results, we go one step further and equate the repeatability to the standard error of the mean which was already depicted in the manuscript. For all the results, we obtained coefficient of variation (COV) less than 10 % for all testing properties.
 - ✓ There are also reasons why the research on laminated veneer products require fewer replicates than other wood samples such as solid wood. One of the advantages of veneer-based wood products is that it is easier to control over the wood property variability and gradients within the final product compared with sawn products (Leggate et al., 2017). By converting log into veneer, defects (knots, cracks and other defects) and natural features (slope of grain) on each veneer are either eliminated during the clipping process or dispersed during production (Pot et al., 2015; H'ng et al., 2012; Tenorio et al., 2011). If defects are left on each veneer sheet, random distribution of such defects during veneer assembly will make the final products more uniform comparable to sawn lumber (Daoui et al., 2011). For instance, in a 15-layer LVL, the effect of a defect in one layer is only 1/15 of what it would be in a piece of solid wood of the respective dimension (H'ng et al., 2012; Anon, 1990). Therefore, veneer-based wood products typically have lower coefficient of variation (COV) in strength and stiffness between 10 and 15 % compared with 25 to 40 % for sawn lumber (Koponen and Kairi, 2002) which the sawn lumber required roughly more than double on the number of replicates.

- ✓ In general, the number of treatment replicates (sample size) depends on the type of materials and experiment you are working on. Mostly in experiments conducted under controlled conditions, your COV should be 20 % or less. Some also accept it even if it is around 30 % especially on research studies where issue of spatial heterogeneity is hard to control for. For all the results, we obtained COV less than 10 % for all testing properties.

Table 1: The value of coefficient of variation (COV), standard deviation, true difference to be detected and calculated value of samples needed for physical properties and bonding quality of LVL produced with veneers peeled from small diameter rubberwood log.

Testing Properties	Highest COV (%)	Standard deviation	True difference to be detected	Number of treatment samples needed (Calculated using Formula 1)
Specific gravity	6.086	0.041	0.123	0.622 ≈ 1
Water absorption after 2 hours	9.814	3.189	7.081	1.136 ≈ 2
Water absorption after 24 hours	7.551	3.862	10.503	0.757 ≈ 1
Radial swelling after 2 hours	9.940	0.144	0.357	0.911 ≈ 1
Radial swelling after 24 hours	6.956	0.259	0.671	0.834 ≈ 1
Longitudinal swelling after 2 hours	9.373	0.024	0.053	1.181 ≈ 2
Longitudinal swelling after 24 hours	9.592	0.083	0.167	1.375 ≈ 2
Tangential swelling after 2 hours	9.711	0.224	0.511	1.075 ≈ 2
Tangential swelling after 24 hours	9.556	0.402	0.960	0.984 ≈ 1
Volumetric swelling after 2 hours	9.708	0.488	1.163	0.987 ≈ 1
Volumetric swelling after 24 hours	6.588	0.624	1.468	1.012 ≈ 2
Gluebond shear strength	7.411	0.287	0.564	1.456 ≈ 2

- ✓ The reviewer question about the need of having 30 replicates. We would like to explain further the differences of “**treatment replicates**” and “**technical replicates**”. In the manuscript we had stated “each treatment was carried out in three replicates” (3 treatment replicates). A **treatment replication** is the repetition of an experimental condition so that the variability associated with the phenomenon can be estimated. ASTM, in standard D906, defines treatment replication as "the repetition of the set of all the treatment combinations to be compared in an experiment". While, **technical replicate** is “when you test the same sample multiple times - repeated measurements of the same sample (the same board) that represent independent measures of the random noise associated with protocols or equipment”. We conducted 3 replications for each treatment (which means 3 boards for each treatment) and 5 - 8 technical replicates for each testing properties on each board (total 15 – 24 technical replicates for each testing properties). All the data of our experiments (results of treatment and technical replications) has been submitted together with the manuscript in the Dryad repository.

REFERENCES:

Juristo N., Moreno A.M. (2001) How Many Times Should an Experiment be Replicated?. In: Basics of Software Engineering Experimentation. Springer, Boston, MA

Leggate, W., McGavin, R. L., Bailleres, H. (2017) A guide to manufacturing rotary veneer and products from small logs. Australian Centre for International Agricultural Research, Canberra, Australia

Pot, G., Denaud, L. E., Collet, R. (2015) Numerical study of the influence of veneer lathe checks on the elastic mechanical properties of laminated veneer lumber (LVL) made of beech. *Holzforschung*, 69(3), 337-345

H'ng, Paik San, Zakiah, A., Paridah, MT. (2012) Laminated Veneer Lumber from Malaysian Tropical Timber: Manufacturing and Design. Penerbit Press, Universiti Teknologi Mara

Tenorio, C., Moya, R., Muñoz, F. (2011) Comparative study on physical and mechanical properties of laminated veneer lumber and plywood panels made of wood from fast-growing *Gmelina arborea* trees. *Journal of wood science*, 57(2), 134-139

Daoui, A., Descamps, C., Marchal, R., Zerizer, A. (2011) Influence of veneer quality on beech LVL mechanical properties. *Maderas Cienc. Tecnol.* 13(1), 69–83

Anon, (1990) A structural Timber Substitute. *Wood Based Panels*. Asia Pacific Forest Industries. Pp 50-53

Koponen, S., Kairi, M. (2002) Structural Composite Lumber. In: *Wood Adhesion and Glued Products: Glued Wood Products State of the Art Report*, Johansson, C.J., Pizzi, T., and Van Leemput, M (eds.), COST Action E13, 69-78.